# Commuting Accidents among Non-Physician Staff of a Large University Hospital Center from 2012 to 2016: A Case-Control Study

**DOI:** 10.3390/ijerph17092982

**Published:** 2020-04-25

**Authors:** Alexandre Ponsin, Emmanuel Fort, Martine Hours, Barbara Charbotel, Marie-Agnès Denis

**Affiliations:** 1Service de médecine et santé au travail, Hospices Civils de Lyon-165 chemin du grand Revoyet, 69495 Pierre Bénite, France; 2Univ Lyon, Univ Eiffel, Univ Lyon 1, Ifsttar, UMRESTTE, UMR T_9405, 69373 Lyon, France; emmanuel.fort@univ-lyon1.fr (E.F.); martine.hours@ifsttar.fr (M.H.); barbara.charbotel@univ-lyon1.fr (B.C.); marie-agnes.denis@chu-lyon.fr (M.-A.D.); 3CRPPE de Lyon, Hospices Civils de Lyon, Centre Hospitalier Lyon Sud, 69310 Pierre-Bénite, France; 4Service de médecine et santé au travail, Hospices Civils de Lyon, 59 Bd Pinel, 69677 Bron cedex, France

**Keywords:** commuting accidents, health care worker, case control study

## Abstract

Road risks (commuting and on-duty accidents) have been responsible for 44% of work-related fatalities compensated by the French system of Social Security in 2012 and still represented 37% in 2018. Our objective was to assess risk factors for commuting accidents among the non-physician staff in a French university hospital. We conducted a case-control study of commuting accidents from 2012 to 2016. Cases were identified and controls were randomly selected from the hospital’s personnel file with matches by year of the accident, gender and age. Risk factors were assessed using conditional logistic regression analysis. An increased risk was observed for 2 × 8 hour shifts, crude OR = 1.40 (95% CI = 1.05–1.86) compared to daytime schedules, but not confirmed in the multiple model. Being a duty officer and not working the day before the accident were associated with increased risk of accidents with adjusted OR = 1.9 (95% CI = 1.1; 3.3) and OR = 1.5, (95% CI = 1.1; 2.1), respectively. The risk increased as the distance between home and work increased, such as adjusted OR = 2.2 (95% CI = 1.4; 3.4) for a distance of >3.6 to 9 km, OR = 2.6, (95% CI = 1.7; 4.0) for a distance of >9 km to 19 km, and OR = 4.2, (95% CI = 2.8; 6.2) for >19 km vs. <3.6 km. The distance between home and work, not working the day before the accident, and certain categories of personnel were related to commuting accidents.

## 1. Introduction

Work-related road-traffic accidents are particularly serious when compared with other work-related accidents [1]. In France, a commuting road-traffic accident is considered to be any road-traffic accident that occurs between employees’ main residence and their workplace, or between their workplace and where they usually take their meals [2].

Data in the French health insurance 2017 annual report showed commuting accidents had increased significantly in almost all regions and were back up to the level observed in 2013 [3]. They represented almost 44% of all fatal occupational accidents recognized under the general social security system in 2012 and 37% in 2018 [4,5]. A study using data from the Rhône Road Accident Victim Register showed an over-representation of health sector professionals among victims of work-related road-traffic accidents, and a significant increase of 16.8% was reported for the health sector at a national level between 2001 and 2014 [6,7]. In 2016, there were 2226 commuting accidents in the French hospital sector, resulting in 68,431 sick leave days, that is, an average of 49.6 sick leave days per injured employee, representing a significant social cost for the individual, the employer and society as a whole [8].

The Hospices Civils de Lyon is one of the largest employers in the Lyon region with some 23,000 employees, including 5000 physicians [9]. In the French hospital civil service, working hours are set at 35 h per week or 1607 h per year [10]. However, some of these employees work atypical hours—They alternate work in 2 × 8-hour shifts (either a morning shift from 6:00 a.m. to 2:00 p.m. or an afternoon shift from 2:00 p.m. to 10:00 p.m.), with the night shift involving 10 h of work. According to the SUMER survey, in 2010, the prevalence of night work in the health sector was 14.5% [11]. In 2012, the proportion of night workers among nurses and midwives was 42%, and for nursing auxiliaries it was 25% [12]. The rhythm of 2 × 12 h shifts, introduced in the 1980s, was extended following the publication of the n°2002-9 decree [13]. This rhythm consists of a maximum of 12 h per working day, followed by a daily rest period of at least 12 consecutive hours and a weekly rest period of at least 36 consecutive hours. Employees cannot work more than three consecutive 12-hour days. The rhythm has been reported to increase satisfaction with working hours, as more time is available for family, social life and domestic activities [14]. In 2014, 14% to 56% of employees in the healthcare sector worked 12-hour shifts [15]. A European survey carried out in 2015 on working conditions showed that 18% of French people worked rotating shifts compared to 21% in Europe, and that 23% of French people worked at least once a month on a night shift compared to 19% in Europe [16]. Finally, in a study on nurses’ working hours in several European countries, a majority of countries had an 8-hour work organisation. However, in the United Kingdom, 32% of nurses declared working 12-hour shifts [17].

Healthcare staff have reported their satisfaction with these modified working hours because they reduce the number of commuting journeys, particularly since many employees live outside the city [18]. Since 1999, employees have become increasingly mobile, both in terms of their work and the distances they travel to work. A study of the average distance travelled in France each year between home and work shows that this distance is steadily increasing. In 2013, more than half of the working population lived more than 15 kilometres from their place of work, 2 kilometres more than in 1999 [19].

In this setting, we performed a case-control study to investigate the occupational factors associated with the risk of commuting accidents, particularly to determine the influence of variable day and night shifts in the non-physician hospital population that often work 8-hour and 12-hour day shifts or 10-hour night shifts. It is important to assess the impact of the modified work schedule on the risk of commuting accidents, since this reorganization represents a financial gain for hospitals and could be further developed [15].

## 2. Material and Methods

We performed a cohort case-control study. Records for permanent and contract hospital non-physician staff, who were employed on 1st January for the years 2012 to 2016 in all the hospital establishments of the University Hospital Center (CHU) were transmitted by the CHU’s information technology department. Commuting accidents were recorded either during an occupational medicine consultation using the occupational health software, Chimed©, or by the statutory medical service after reception of a medical certificate and the administrative declaration. The occupational physician has access to information about all accidents, which guaranteed completeness.

### 2.1. Study Population

#### 2.1.1. Cases

Information on commuting accidents from 2012 to 2016 were extracted from the medical service personnel file using the occupational health software, Chimed©. The inclusion criteria for cases were non-physician hospital staff working in CHU hospitals in the Lyon region that had a reported commuting accident between 2012 and 2016. The data set was screened to eliminate accidents involving physicians, because they have a different employment status and data are not systematically collected for them, as well as to check the commuting accidents were correctly coded and that they were not occupational accidents. Commuting accidents were classified by type, that is, not involving a vehicle, involving a car, involving a motorized two-wheeler vehicle or a bicycle, or pedestrian run over by a vehicle. If a case had more than one community accident, only the first was included in our analyses. Among the 390 commuting accidents identified in 390 cases, 17 had insufficient usable information and 15 involved staff working in an establishment dependent on the University Hospital located in another department and therefore considered to be outside the study area, leaving 358 cases that were analyzed.

#### 2.1.2. Controls

Four age- and gender-matched controls were randomly identified for each case from the personnel file corresponding to the year the accident occurred, using the SAS SURVEYSELECT procedure without replacement, and a macro developed under SAS. This macro selected the four matched controls and had been used for the five databases of each year of the study. Information on work schedules was not available for 280 (18%) of the 1560 identified controls, leaving 1280 controls that were included in the analyses.

### 2.2. Data Collected

Employees’ personnel numbers were used to find their 12-month working time history in the personnel file. The controls had to be scheduled to work on the day that the accident occurred for the case to which they were matched; this date was taken as their reference date.

The information extracted from the personnel file was:The day of the accident (reference date): Start and end time of shift, time worked for the day, type of day—that is, weekday, weekend day, a public holiday;Worked the day before the accident, and if yes, what time;Total number of hours and number of night hours worked during the seven days before the accident;Number of rest days during the seven days before the accident;Work rhythm: 2 × 8 h, 2 × 12 h, night or day.

The distance from home to work was calculated using Google Maps© with the employee’s home and workplace addresses without the coder knowing the case or control status.

### 2.3. Professional Classification

The positions of the cases and controls were defined using nine professional categories:Administrative personnel;Service personnel;Nurse-anesthetists, operating room nurses and midwives;Laboratory or pharmacy personnel and their supervisors;Socio-educators, facilitators, dieticians, occupational therapists, physiotherapists, electro-radiology manipulators and all paramedical staff in contact with patients, and their supervisors;Auxiliary nurses and childcare assistants;Nursing executives, specialist nursing executives, midwife coordinators;Nurses;Technical personnel.

### 2.4. Statistical Analysis

Studied variables:

The principal outcome was defined as having a commuting accident or not. The exposure factors were:Work rhythm (2 × 8 h, 2 × 12 h, night or day);Professional category;Worked the day before the accident;Worked during the seven days before the accident:
○Number of rest days○Number of hours worked○Number of night hours workedNumber of days worked since last rest day;Home-to-work distance categorized into quartiles using data for controls (<3.6 km, 3.6 to 9 km, 9 to 19 km, and >19 km).

Analysis schedule:

As a first step, univariate analyses were performed to calculate the non-adjusted odds ratio. In the second step, a conditional logistic regression multiple analysis including the different exposure factors of interest was performed using the LOGISTIC procedure in SAS 9.4. The selection of variables was a backward stepwise selection with an entry threshold of 10% and an exit threshold of 5%. After each iteration, the possible confounding effects of the output variable of the model were checked. A 10% deviation of a parameter was considered to be a confounding-induced change.

### 2.5. Regulatory and Ethical Considerations

The study Chimed© file was declared and registered in the CHU’s CNIL register under number 07-3. Following a request from The CHU’s data protection and compliance manager, we approved the project because personal data was accessible only by the occupational physician. This project was also approved by the CHU’s ethics committee. The project did not require approval from the independent protection committee since it was a retrospective study using existing data and was therefore outside the scope of law no. 2012-300 of 5 March 2012, known as the Jardé law [20].

## 3. Results

### 3.1. Characteristics of Cases

The study population included 358 cases and 1280 controls, with an average of 3.58 controls per case (median = 4, minimum = 2, maximum = 4). The average age of the cases was 42 years (standard deviation (SD) = 11.5 years) with a range of 20 to 64 years, and 80% were women (Table 1). Sixty-eight percent had a partner and 23% were single.

The population had worked an average of about 30 h during the seven days before the accident. The percentage of accidents was homogeneous over the weekdays with about 17% per day and half as many on weekends. A total of 79% of the cases had sick leave following the commuting accident for an average of 50 days. The accident occurred on the way to work for 71% of the cases (69% among day workers, 83% among 12-hour shifters, 62.4% among 2 × 8 shifters and 100% for night shifters, respectively). The most frequent professional category was nurses (26.8%) auxiliary nurses and childcare assistants (25.7%), administrative staff (13.1%) of service personnel (11.5%), and technical personnel (9.5%) (Table 1). Auxiliary nurses and childcare assistants and nurses represented about 25% of those living more than 19 km from their place of work. Nearly 50% of those who worked during the day lived >19 km from work and about 25% each of those who worked 12-hour and 2 × 8 hour.

### 3.2. Univariate Analyses

An increased risk of commuting accidents was observed for service personnel compared with administrative personnel (OR = 2.0, 95% CI: 1.2; 3.3). An increased risk was observed for 2 × 8 hour shifts, crude OR = 1.40 (95% CI = 1.05–1.86) compared to daytime schedules, but not for other schedules.

The risk of having a commuting accident increased with the distance from the workplace: OR = 2.2, (95% CI: 1.5; 3.4) for a distance of between 3.6 and 9 km; OR = 2.6, (95% CI: 1.7; 3.9) for a distance between 9 km and 19 km; and OR = 4.3, (95% CI: 4.3; 6.4) for a distance >19 km compared with those who lived <3.6 km from their workplace.

Not working the day before was associated with a higher risk of having a commuting accident (OR = 1.8, 95% CI: 1.4; 2.5). Working more than 39 h during the seven days before the reference date was associated with a lower risk of accidents (OR = 0.6, 95% CI: 0.4; 0.9) (Table 2).

### 3.3. Multivariate Analyses

Service personnel and not working the day before remained significantly associated with a higher risk of commuting accidents in multivariate analyses, OR = 1.9 (95% CI: 1.1; 3.3) and OR = 1.5 (95% CI: 1.1; 2.1), respectively (Table 3). Longer distances between home and work also remained significantly associated with a higher risk of commuting accident distances, OR = 2.2 (95% CI: 1.4; 3.4) for between 3.6 km and 9 km, 2.6, (95% CI: 1.7; 4.0) for between 9 km and 19 km, and 4.2, (95% CI: 2.8; 6.2) for >19 km compared with those who travelled less than 3.6 km, with no significant dose-response relationship found.

## 4. Discussion

Our results showed that service personnel had a higher risk of commuting accidents in multivariate analysis, OR = 1.9 (95% CI: 1.1; 3.3). These workers have more physical constraints (e.g., manual or patient handling, postural strain to make beds, physical aggression, prolonged standing) than the others and are therefore likely to be more fatigued, at least partly due to the atypical work schedules they have and the need to be frequently standing to do their job. Several studies have also reported a possible association between road traffic accidents and fatigue, lack of sleep, a high mental workload at work and work rhythms [21,22,23,24,25,26]. It has been suggested that these same factors could also explain the excess risk observed among healthcare professionals in the United States [27,28,29,30].

When we analyzed the atypical work shifts, using daytime work as the reference, there was no significant increased risk of commuting accidents for those who worked 2 × 12 h, 2 × 8 h or night shifts from the multiple analyses. More than 70% of the cases had an accident on the way to work, which suggests that the cumulative fatigue from the working day does not appear to be a risk factor, as described in a previous study [6]. On the contrary, stress related to being on time at work can be a risk factor. We did not observe any higher risk of commuting accidents on the way to work before a morning shift nor on the way home after a night shift, as reported in other studies [31]. One study showed that working long hours necessitates staying awake for longer periods of time and slows cognitive function and reaction time to a level equivalent to that associated with a blood alcohol concentration of 0.5 g/L [32]. One systematic literature review reported that working longer than 8 h results in an increased risk of accidents; hence, working for 12 h doubles the risk of accidents compared to the risk after working 8 h [33]. It was calculated that the risk of road traffic accidents is 10 times higher at night than during the day, after adjusting for traffic intensity [34]. An expert report published by the French Agency for Food, Environmental and Occupational Health and Safety (ANSES) in 2016 stated that atypical working hours can lead to an increase in the frequency and severity of road traffic accidents, with a doubling of the risk of road traffic accidents and “near accidents” due to sleep disturbance [35]. A review published in 1997 concluded that the risk for road traffic accidents follows a pronounced circadian rhythm, while a more recent study suggested that, although circadian rhythms do play a role, there are confounding factors that require further research to clarify the link [36,37].

Furthermore, the risk of road traffic accidents increases 0.6% for each additional km travelled by employees from home to work. The results from our multivariate analysis confirmed this trend. In our population, half of the employees working atypical shifts were in the furthest quartile (>19 km), which was a comparable proportion to those working daytime hours. Vigilance must be maintained when travelling, especially when home is far from the workplace, although one study reported that more than half of all commuting accidents occur within 5 km of the workplace [6]. The French health insurance report stated that the increase in commuting accidents recorded in 2018, compared with those recorded in 2017, could also be explained by weather conditions, with more accidents occurring in winter and in regions most exposed to bad weather [2].

Not working the day before the road traffic accident was significantly associated with a higher risk of commuting accidents, OR = 1.8 (95% CI: 1.4; 2.5), which suggests that the cumulated working hours did not explain the increased risk. It is possible that difficulties to adjust to changing rhythms could explain the increased risk. These results are consistent with data from the 2017 Hospital Public Service Statistical Report indicating that commuting and work-related accidents were more frequent at the beginning of the working week [8].

Although all the commuting accidents experienced by our cases were declared to the occupational medical service, it is possible that some employees did not declare their road traffic accident as an occupational accident. There was no measurement bias in our study since the information of interest was obtained from the personnel file in the same way for cases and controls, without the employees having to fill in a questionnaire individually.

Based on these results, several preventive measures could be implemented. The most obvious one is the reinforcement of information for workers on road risks by occupational health services, but also by other means, such as flyers or conferences. They should particularly be concerned with things such as compliance with rules on alcohol consumption, speed limits, seatbelt wearing, use of mobile phones while driving, as well as having a healthy lifestyle (i.e., sleep, physical activity, diet). This prevention could more specifically target the most exposed occupational categories, as well as people travelling long distances to and from work. It would be preferable to improve access to the public, rather than individual means of transport by increasing the financial coverage of public transport season tickets and facilitating the development of car-pooling, as already suggested by the CNAMTS (the French national health insurance fund for salaried employees) [38].

## 5. Conclusions

Working as service personnel, living far from the workplace and not working the day before the road traffic accident were significantly associated with a higher risk of commuting accidents. Prevention programs focusing on road traffic risks and rules for healthy living and sleep for the most exposed occupational categories, as well as on staff who live the furthest from their workplace could be beneficial.

## Figures and Tables

**Table 1 ijerph-17-02982-t001:** Characteristics of cases included in the study.

Variables	Modalities	N	Percentage (or Mean (SD))
Gender	Female	284	79.33
	Male	74	20.67
Age (years)		358	41.57 (11.46)
Marital status	With a partner	224	67.68
	Single	76	22.96
	Divorced, widowed or separated	31	9.36
Type of commuting accident	Involving a car	285	79.6
	Not involving a vehicle	16	4.5
	Involving a motorized two-wheeler vehicle or a bicycle	57	15.9
Sick leave	No	76	21.23
	Yes	282	78.77
Duration of sick leave (days)		358	39.75 (105.06) (range [0d-1271d], median= 10d)
Duration of sick leave among those on sick leave (days)		282	50.46 (116.73) (range [1d-1271d], median= 17d)
Number of accidents	1	209	58.38
	2	72	20.11
	3	27	7.54
	4	23	6.42
	≥6	27	7.54
Day accident occurred	Monday	58	16.20
	Tuesday	69	19.27
	Wednesday	53	14.80
	Thursday	61	17.04
	Friday	64	17.88
	Saturday	24	6.70
	Sunday	29	8.10
Year accident occurred	2012	81	22.63
	2013	69	19.27
	2014	64	17.88
	2015	55	15.36
	2016	89	24.86
Type of journey	Home to work	255	71.2
	Work to home	103	28.8
Professional category	Administrative personnel	47	13.13
	Service personnel	41	11.45
	Nurse-anesthetists, operating room nurses and midwives	8	2.23
	Laboratory or pharmacy personnel and their supervisors	18	5.03
	Socio-educators, facilitators, dieticians, occupational therapists, physiotherapists, electro-radiology manipulators and all paramedical staff in contact with patients, and their supervisors	17	4.75
	Auxiliary nurses and childcare assistants	92	25.70
	Nursing executives, specialist nursing executives, Midwife coordinators	5	1.40
	Nurses	96	26.82
	Technical personnel	34	9.50

**Table 2 ijerph-17-02982-t002:** Univariate analyses of commuting accidents from 2012 to 2016 in cases and controls drawn from non-physician staff working in a large teaching hospital center.

Variables	Modalities	Cases		Controls		OR	95% CI	*p*-Value (Type 3 Test)
Professional category (n, %)	Administrative personnel	47	13.1	179	14.0	1	-	0.02
	Service personnel	41	11.5	87	6.8	1.96	1.17–3.28	
	Nurse-anesthetists, operating room nurses and midwives	8	2.2	37	2.9	0.9	0.39–2.04	
	Laboratory or pharmacy personnel and their supervisors	18	5.0	56	4.4	1.3	0.70–2.42	
	Socio-educators, facilitators, dieticians, occupational therapists, physiotherapists, electro-radiology manipulators and all paramedical staff in contact with patients, and their supervisors	17	4.8	93	7.3	0.73	0.40–1.32	
	Auxiliary nurses and childcare assistants	92	25.7	265	20.7	1.38	0.92–2.05	
	Nursing executives, specialist nursing executives, Midwife coordinators	5	1.4	29	2.3	0.69	0.26–1.89	
	Nurses	96	26.8	391	30.5	0.68	0.66–1.45	
	Technical personnel	34	9.5	143	11.1	0.96	0.57–1.62	
Distance (n, %)	<3.6 km	35	9.86	319	25.08	1	-	<0.0001
	3.6 to 9 km	79	22.25	318	25	2.23	1.45–3.44	
	9 to 19 km	91	25.63	316	24.84	2.55	1.67–3.88	
	>19 km	150	42.25	319	25.08	4.25	4.25–6.36	
Worked the day before the accident (n, %)	No	133	37.2	368	28.8	1.8	1.4–2.5	0.0009
	Yes	225	62.8	912	71.2	1	-	
Work rhythm (n, %)	Daytime	177	49.4	687	53.7	1	-	0.09
	2 × 12 h	59	16.5	223	17.4	1.03	0.74–1.44	
	2 × 8 h	104	29.1	289	22.6	1.40	1.05–1.86	
	Night-time	18	5.0	81	6.3	0.88	0.51–1.50	
Time range the day before the accident (n, %)	Rest day	133	37.2	368	28.7	1.83	1.35–2.50	0.003
	Afternoon	28	7.8	85	6.6	1.57	0.98–2.52	
	Day	120	33.5	550	43.0	1	-	
	Morning	50	14.0	170	13.3	1.36	0.94–1.97	
	Night	27	7.5	107	8.4	1.17	0.74–1.86	
Number of rest days during 7 days before the accident (n, %)	1	5	1.4	37	2.9	0.53	0.20–1.39	0.06
	2	111	31.0	434	33.9	1	-	
	3	89	24.9	324	25.3	1.09	0.80–1.51	
	4	88	24.6	225	17.6	1.56	1.12–2.27	
	5	37	10.3	163	12.7	0.87	0.57–1.33	
	6	16	4.5	56	4.4	1.06	0.58–1.93	
	7	12	3.3	41	3.2	1.09	0.55–2.15	
Number of hours worked the 7 days before the accident (n, %)	<25 h	101	28.2	323	25.2	1	-	0.01
	25 h to 35 h	113	31.6	394	30.8	0.92	0.67–1.25	
	35 h to 39 h	85	23.7	249	19.5	1.09	0.78–1.53	
	>39 h	59	16.5	314	24.5	0.61	0.43–0.88	
Number of night hours worked the 7 days before the accident (n, %)	None	302	84.4	1079	84.3	1	-	0.9
	<24 h	26	7.3	101	7.9	0.91	0.58–1.44	
	≥24 h	30	8.4	100	7.8	1.09	0.71–1.66	
Number of working days since last rest day (n, %)	1	46	12.8	185	14.5	1	-	0.2
	2	105	29.3	318	24.8	1.31	0.89–1.95	
	3	71	19.8	230	18.0	1.26	0.83–1.92	
	4	54	15.1	186	14.5	1.16	0.75–1.81	
	5	82	22.9	361	28.2	0.89	0.60–1.34	
Distance between home and work (mean, SD)		21.65	27.58	16.04	27.13	1.006	1.00–1.01	0.002
Number of days not worked during the 7 days before the accident		Mean = 3.38 SD = 1.35		Mean = 3.29 SD = 1.4		1.04	0.96–1.14	0.3
Number of hours worked during the 7 days before the accident (mean, SD)		Mean = 30.2 SD = 10.29		Mean = 31.23 SD = 10.87		0.99	0.98–1.01	0.12
Number of days worked since last rest day (mean, SD)		Mean = 3.06 SD = 1.37		Mean = 3.17 SD = 1.44		0.94	0.87–1.03	0.2

**Table 3 ijerph-17-02982-t003:** Multivariate analyses of commuting accidents from 2012 to 2016 in cases and controls drawn from non-physician staff working in a large teaching hospital center.

Variables	Modalities	OR	95% CI	*p*-Value (Type 3 Test)
Professional category	Administrative personnel	1.0	-	0.02
	Service personnel	1.9	1.11–3.25	
	Nurse-anesthetists, operating room nurses and midwives	0.82	0.35–1.91	
	Laboratory or pharmacy personnel and their supervisors	1.18	0.63–2.24	
	Socio-educators, facilitators, dieticians, occupational therapists, physiotherapists, electro-radiology manipulators and all paramedical staff in contact with patients, and their supervisors	0.6	0.33–1.12	
	Auxiliary nurses and childcare assistants	1.25	0.83–1.89	
	Nursing executives, specialist nursing executives, Midwife coordinators	0.56	0.20–1.56	
	Nurses	0.93	0.62–1.40	
	Technical personnel	0.90	0.50–1.60	
Distance between home and work	<3.6 km	1	-	< 0.0001
	3.6 to 9 km	2.2	1.41–3.41	
	9 to 19 km	2.6	1.7–4	
	>19 km	4.15	2.75–6.24	
Worked the day before the accident	No	1.55	1.15–2.08	0.005
	Yes	1	-	
Number of hours worked in the 7 days before the accident	<25 h	1	-	0.04
	25 h to 35 h	1.4	0.9–2.0	
	35 h to 39 h	1.8	1.2–2.6	
	>39 h	1.3	0.9–2.0	

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
