# Peer review of "Commuting Accidents among Non-Physician Staff of a Large University Hospital Center from 2012 to 2016: A Case-Control Study"

_ijerph, 2020, doi:10.3390/ijerph17092982_

Round 1

Reviewer 1 Report

This paper analyzed the data of commuting accidents among non-physician staff serviced at a large university hospital center. Data was collected from 2012 to 2016. The results showed that working as service personnel, living far from the workplace, and not working the day before the road traffic accident have higher risks. This paper has some shortcomings as follow:

  1. In the section of univariate analyses, thirteen variables are analyzed. However, in the multivariate analysis, only, four variables are analyzed. The authors cannot explain why the adopted variables are different between univariate analysis and multivariate analysis.
  2. The authors found some variables will result in commuting accidents. Hence, the preventing strategies for decrease commuting accidents should be suggested. Or, the revised occupational safety and health strategies could be explored. Hence, this paper should have a section to explore the strategy in order to decrease the occurrence of commuting accidents.

Reviewer 2 Report

Pertinent and concise abstract, yet when providing figures in the abstract I would suggest to either add a reference or giving a general trend

Motivations for this research are well contextualised in the literature review with key facts and trends; yet what’s the proportion of the night/day shifters at focus? How do French average numbers (distances, hours worked etc. in this sector) compare with European and international ones?

In the methodology if combining SAS SURVEY SELECT and a SAS macro development is a standard approach, the authors should mention it for clarity

I am not directly familiar with the dose effect functions, but the research methodology seems well conducted.

The results interpretation could be more widely discussed; what to conclude from the fact that not working on the previous day increase the fatality risk? Which policy/managerial recommendations?

Reviewer 3 Report

General comments.

I think this paper is very well elaborated and might be of interest not only for the local but also for the global Road Traffic Injury (RTI) prevention community. In my opinion, findings have the potential to inform stakeholders so they can improve road safety at the local and national levels while they might also be of interest for the RTI prevention community (as this work is in line with the new sustainable development agenda at least in the goals 3.6 and 8.8).

The paper seeks to assess risk factors for commuting accidents among non-physician staff in a French university hospital, using a case-control study design. As the authors clearly mention, work related injuries in general, and commuting accidents in particular, have increased recently in France. This might be the case for other countries such as LMICs.

While the term is not necessarily incorrect, I would challenge authors not to use the word “accident”. This term has the connotation of being “unavoidable” (which is not the case for any RTI), and might be confusing as sometimes is used to refer to the crash or collision and sometimes to the health consequences (fatal and non-fatal RTI). It is not always clear which is meant.

I have some comments and questions that I hope could contribute to improve the document. I present them by section:

Introduction

  • Authors mention that distance between home and work is steadily increasing. To strengthen their point, I suggest they include data from another year to compare the 2008 data on average km or time in minutes per one-way journey that they present in rows 55-56.
  • A little more context information about the week shifts would be useful for international readership. For example, 2 x 8h means what exactly?, and how many hours per week (in average) all these schemes imply? I believe this is relevant as the objective of the paper is “to evaluate occupational factors associated with the risk of commuting accidents, in particular to determine the influence of variable day and night shifts…”

Material and Methods:

  • Were cyclists traffic collisions/injuries included? If not, I would recommend authors to explain why not.
  • While only the first traffic-related commuting injury was considered to define a “case”, 42% of the cases had two or more commuting events; 7.54% suffered six or more. Would it be worth to conduct a more in depth analysis with this sub-population? One can argue that they might have some specific characteristics that might provide interesting results. Is there a way to compare the severity of injuries between different events?
  • Cases:2% of the cases were lost (17 had insufficient usable information and 15 involved staff working in an establishment dependent on the University Hospital located in another department and therefore considered to be outside the study area). Were eliminated cases different in other relevant variables as the cases that were included in the study sample?
  • Controls: Authors should present more information to allow potential readers to assess the extend to which excluding the 280 individuals with missing information on work schedules (17.9% of the 1,560 age and gender matched controls randomly identified) might have biased the results of the study. Were these 280 individuals different in other relevant variables as those who were finally included as controls?
  • Would it be better to say “exposure factors/variables” instead of “secondary outcomes” (rows 123-124?
  • What was the rationale behind grouping distance from home to work in those specific categories (i.e. <3.6km, 3.6 to 9km, 9 to 19km and >19)?
  • It would be desirable that authors specify how they evaluated the goodness of fit of the multiple (not really multivariate, which refers to multiple response variables) logistic regression analysis. Some of this information might be included as footnotes in table 3 for interested readers.

Results:

  • Besides average of days of sick leave following the traffic-related commuting injuries, I would recommend authors including min, max and median.
  • Table 2 says univariate but in essence, it is actually a bivariate analysis (i.e. distribution of each independent covariable of interest by case/control status). I suggest authors present all P-values for each category of comparison and not only those statistically significant (this recommendation applies also for table 3). This will improve clarity as, for example, p-value of 0.02 in the professional category corresponds to the category “service personnel” but is currently aligned to the category “Socio-educators…” which in fact is not statistically significant (OR=0.73, 95% CI: 0.40-1.32).
  • I think analysis by sex and age group is missing from table 2 and table 3. It is somehow hard to believe that none of these two variables was not associated to commuting accidents or that models fitted well without including these two well-known confounding variables. Sometimes these two variables interact with each other; was any relevant interaction such as this and other (i.e. interaction between work schedule and distance between home/work) tested?

Discussion:

  • I think there is no need to repeat results in the initial two sentences of first paragraph of the discussion section. Instead, authors should summarize key results in reference with their study objective(s).
  • What exactly do authors mean by “more physical constraints” (row 189)?
  • Authors mention that “Several studies have also reported a possible association between road traffic collisions and fatigue, lack of sleep, high mental workload at work and work rhythms”. But they did not explore any of this and thus it should be stated as a potential limitation of their study.
  • It would be interesting for potential readers if authors can further develop ideas/hypothesis as to why:
    • … commuting accidents occurred on the way to work more frequently than on the way home?
    • … commuting accidents increased in 2016 after the decreasing trend observed from 2012 to 2015?
    • … working more than 39 hours during the seven day before the reference date was not associated with a higher risk of having commuting accidents as compared to those working less than 25 hours?
    • … work rhythm was not associated to the risk of having commuting accidents?

Round 2

Reviewer 1 Report

Accept in present form